# Phenotypic and Genotypic Characterization and Antifungal Susceptibility of *Sporothrix schenckii sensu stricto* Isolated from a Feline Sporotrichosis Outbreak in Bangkok, Thailand

**DOI:** 10.3390/jof9050590

**Published:** 2023-05-18

**Authors:** Kanokporn Yingchanakiat, Orawan Limsivilai, Supita Sunpongsri, Waree Niyomtham, Kittitat Lugsomya, Chompoonek Yurayart

**Affiliations:** 1Department of Microbiology and Immunology, Faculty of Veterinary Medicine, Kasetsart University, 50 Ngamwongwan Road, Bangkok 10900, Thailand; kanokporn.yin@ku.th (K.Y.); fvetorw@ku.ac.th (O.L.); 2Veterinary Teaching Hospital, Faculty of Veterinary Medicine, Kasetsart University, 50 Ngamwongwan Road, Bangkok 10900, Thailand; supita.su@ku.th; 3Department of Veterinary Microbiology, Faculty of Veterinary Science, Chulalongkorn University, Henri-Dunant Road, Pathumwan, Bangkok 10330, Thailand; pewwaree@yahoo.com; 4Antimicrobial Resistance and Infectious Diseases Laboratory, Harry Butler Institute, Murdoch University, 90, South Street, Murdoch, Perth 6150, Australia; kittitatlugsomya08@gmail.com

**Keywords:** sporotrichosis, *Sporothrix schenckii*, cats, phenotypic characterization, genotypic characterization, antifungal susceptibility

## Abstract

Sporotrichosis, an invasive fungal infection caused by *Sporothrix schenckii*, has emerged in Southeast Asia, affecting cats and posing a potential zoonotic risk to humans. We evaluated 38 feline sporotrichosis cases in and around Bangkok, Thailand, from 2017 to 2021. The isolates were phenotypically and genotypically characterized. The cats infected with sporotrichosis were mainly young adults, males, and domestic short hairs with uncontrolled outdoor access, and they lived in Bangkok. All isolates showed low thermotolerance and converted to the yeast phase at 35 °C. Based on the internal transcribed spacer region of rDNA sequences, our strains belonged to *S. schenckii sensu stricto* and clustered with clinical clade D. Based on the concatenated tree of calmodulin and beta-tubulin genes, five groups of *S. schenckii* were generated, and the monophyletic clade, Group II, of Thai strains was recognized. In vitro antifungal susceptibility testing demonstrated that the MIC50 of our isolates to amphotericin B, itraconazole, and posaconazole were within the limit of the species-specific epidemiological cutoff values, suggesting that the organisms were the wild type. Addressing the outbreak of feline sporotrichosis in Thailand by providing guidelines for diagnosis and effective treatment may help control the spread of disease and reduce the risk of cat-transmitted sporotrichosis to humans.

## 1. Introduction

Sporotrichosis is a subcutaneous and systemic mycotic infection that affects both humans and animals. It is caused by pathogenic dimorphic fungi of the genus *Sporothrix*, which is plentiful in soil and plant debris. Sporadic infection through skin trauma during activities involving contact with vegetation and soil, such as gardening, farming, or mining among humans, or digging and clawing among cats, has been reported globally [1]. In endemic areas of human and feline sporotrichosis, such as Latin America (Brazil, Colombia, Peru, Argentina, and Venezuela) and Southeast Asia (Malaysia), sporotrichosis has become an emerging health problem from epizootic (cat–cat or cat–dog) and zoonotic (cat–human) infections [2].

Cats with sporotrichosis may exhibit a wide range of clinical signs, from no symptoms to fetal disseminated systemic disease. The most common lesions observed are skin ulcers, granulomatous nodules, and crusts at various sites on the body, particularly on the head, nose, ears, distal limbs, and tail base. Lesions in the nasal passages may make the infection difficult to treat and can lead to extracutaneous signs, such as sneezing, rhinorrhea, and dyspnea, which can lead to therapeutic failure and death [3].

Early diagnosis of sporotrichosis usually relies on direct microscopy of budding yeasts with cigar-shaped bodies from suspected lesions. However, misdiagnosis often occurs, especially among other organisms with yeast-like morphology, such as *Histoplasma capsulatum* and *Cryptococcus neoformans*. The gold standard for sporotrichosis diagnosis is fungal isolation and identification; this approach can inform proper treatment and provide useful data for reducing clinical loss and the risk of sporotrichosis outbreaks in humans and animals [4].

*S. schenckii* and related species are the most prevalent agents causing worldwide invasive zoonotic and epizootic infections in humans and animals [5,6]. Genotypes of *Sporothrix* spp. are characterized by various genetic markers; the internal transcribed spacer (ITS) region of rDNA, calmodulin (CAL), beta-tubulin (b-TUB), and translation elongation factor-1 alpha (TEF1a) inform the epidemiological distribution associated with geographic areas [7]. *S. brasiliensis* is emerging in Brazil, *S. schenckii sensu stricto* (*S. schenckii s. str.*) and *S. globosa* are distributed in many continents, and *S. luriei* is found in Africa, Italy, and India [8,9,10]. Currently, the presence of feline-related sporotrichosis is increasing in Asia. A study of the outbreak in Malaysia revealed a single clonal strain of clinical clade D, *S. schenckii s. str.*, as the predominant trait in the environment causing human and feline sporotrichosis there [11].

The first feline sporotrichosis case in Thailand was described in 2017 and was followed by a few Thai cases of sporotrichosis in humans with no history of previous trauma or contact with an affected animal. Therefore, promoting knowledge about feline sporotrichosis in Thailand is urgent. In this study, we collected and summarized clinical data on feline sporotrichosis; characterized the phenotype and genotype of *S. schenckii* isolated from a feline sporotrichosis outbreak in Bangkok, Thailand, and the surrounding regions during 2017–2021; and assessed the susceptibility of the organisms to commonly used antifungal drugs.

## 2. Materials and Methods

### 2.1. Fungal Isolates and Data Collection

Forty-two isolates from cats (n = 38) and a cat owner (n = 1) with skin lesions suspected of sporotrichosis were obtained from the Veterinary Teaching Hospital at Kasetsart University, Bangkok, Thailand, and isolates from untraceable source data (n = 3) were obtained from a private veterinary diagnostic laboratory in Bangkok. All isolates were recovered between December 2017 and December 2021.

Informed consent was obtained from the cat owners before veterinarians performed direct examination by impression smears and sample collection by wound swabbing. Samples were sent for fungal culture and identification to Microbiology Testing Services in the Department of Microbiology and Immunology, Faculty of Veterinary Medicine, Kasetsart University (registered project number: 00751/63). Samples were then inoculated on Sabouraud dextrose agar (SDA) and brain heart infusion agar (BHIA) containing chloramphenicol (100 mg/L) and amoxicillin (50 mg/L) and incubated at 30 and 35 °C for 5–14 days. The causative agent was initially identified based on dimorphic characteristics of macroscopic and microscopic features of *Sporothrix* spp. All isolates were stored in sterile water as fungal culture collections for further study.

Demographic and clinical data on the 38 feline sporotrichosis cases were recorded, including geographic origin, age, sex, breed, castration, cat rearing, immunodeficiency disorder, possible route of transmission, number of lesions, type of lesion, treatment, and outcome.

### 2.2. Phenotypic Characterization

We observed the growth of 26 isolates of *S. schenckii* on various media (SDA, potato dextrose agar [PDA], and BHIA) and at various temperatures (30, 35, 37, and 40 °C). The growth and transitions to the mycelial and yeast phases, influenced by the culture medium and temperature, were closely monitored under a stereomicroscope (SZ2-ILST, Olympus) for 14 days. The colony diameter was measured by calibrating the ocular micrometer on the imaging software (cellSens V3.2 software, Olympus). The microscopic features of the mold form were evaluated using the slide culture technique on PDA, incubated at 30 °C for 14 days. Morphologies of the yeast-like and yeast forms were obtained from culturing on BHIA at 30 and 35 °C for 14 days. The slides were examined under a microscope (BX43, Olympus), and the cells were measured by a calibrated ocular micrometer on the imaging software (cellSens V3.2, Olympus). The percent growth inhibition (% GI) was calculated using the following formula: [(average colony diameter at low temperature-average colony diameter at high temperature)/average colony diameter at low temperature] × 100. Isolates exhibiting more than 50% GI were classified as having low thermotolerance, whereas those showing less than 50% GI were classified as having high thermotolerance [12].

### 2.3. Genotypic Characterization

The genomic DNA of 26 isolates was extracted by grinding with a pestle in liquid nitrogen, followed by the Ferrer et al. [13] extraction method. The ITS region was amplified with primers SR6R (5’-AAGTATAAGTCGTAACAAGG-3’) and ITS4 (5’-TCCTCCGCTTATTGATATGC-3’) [14], the CAL gene was amplified with primers CL1 (5’-GA(GA)T(AT)CAAGGAGGCCTTCTC-3’) and CL2A (5’-TTTTTGCATCATGAGTTGGAC-3’) [15], and the b-TUB gene was amplified with primers BT2-F (5’-GGTAACCAAATCGGTGCTGCTTTC-3’) and BT2-R (5’-ACCCTCAGTGTAGTGACCCTTGGC-3’) [16]. The amplicons were sequenced using the capillary electrophoresis sequencing method, using ABI Bigdye Terminator v. 3.1 (Macrogen Inc., Seoul, South Korea). The sequences generated in both forward and reverse directions were assembled and manually corrected for consensus sequences using BioEdit software v. 7.0.5.3. Species of all isolates were predicted using the BLASTn tool of the National Center for Biotechnology Information (https://blast.ncbi.nlm.nih.gov/Blast.cgi, accessed on 3 September 2020 for isolates recovered during 2017–2020, and on 15 August 2021 for isolates recovered in 2021).

The ITS phylogenetic analysis was performed using MEGA11 software for *Sporothrix* species identification [17]. The phylogenetic trees were constructed using the maximum likelihood (ML) method, the reliability of each node was assessed using a bootstrap analysis of 1000 replicates, and the evolutionary distances were computed using the Kamura 2-parameter method [18,19]. The concatenated CAL and b-TUB genes were aligned by MEGA11, and the phylogenetic tree was generated using the ML method with the GTR+Γ substitution model and 1000 bootstrap replicates in the IQ-TREE 2 [20]. The tree was visualized with iTOL online [21]. All sequences in this study were analyzed against the reference sequences from the previous studies presented in Table 1. In brief, 165 sequences were retrieved from nucleotide sequence databases (GenBank): 135 sequences of *S. schenckii* (49 human, 70 cat, 3 environment, and 13 undefined origin), 20 sequences of *S. globosa* (11 human, 9 undefined origin), 1 sequence of *S. luriei* (human origin), 7 sequences of *S. brasiliensis* (4 human and 3 undefined origin), 1 sequence of *S. pallida* (environment origin), and 1 sequence of *S. chilensis* (human origin) [22,23,24,25,26,27].

### 2.4. Antifungal Susceptibility Test

Twenty-six isolates of *S. schenckii* were tested for their susceptibility to eight antifungal agents: amphotericin B (AMB), itraconazole (ITR), posaconazole (POS), ketoconazole (KET), fluconazole (FLU), miconazole (MCZ), terbinafine (TERB), and griseofulvin (GRI). The minimum inhibitory concentration (MIC) and minimum fungicidal concentration (MFC) were determined using the broth microdilution (BMD) method according to the Clinical and Laboratory Standards Institute (CLSI M38-A2) [28]. Antifungal agents were dissolved according to CLSI recommendations and diluted in the RPMI 1640 (Invitrogen, Waltham, MA, USA) buffered to a pH of 7.0 with 0.65 M of MOPS buffer (Sigma, St. Louis, MO, USA) with L-glutamine and without bicarbonate to obtain 2X concentrations. They were loaded to 96-well U-shaped microplates at a final concentration of 0.06 to 32 µg/mL for AMB, ITR, POS, and GRI; 0.0156 to 8 µg/mL for KET, MCZ, and TERB; and 0.25 to 256 µg/mL for FLU (all were purchased from Sigma, St. Louis, MO, USA). Antifungal susceptibilities of both the mold and yeast phases were tested by harvesting the cells from 3-day-old cultures growing on PDA at 30 °C and BHIA at 35 °C, respectively. The conidia and yeast cells were scraped from the colony surface using a sterile cotton swab and stirred into a sterile saline solution. The inoculum suspension was vortexed and allowed to settle for 5 min, and the upper homogeneous layer was transferred, measured, and adjusted to an optical density range of 0.09 to 0.11 at 530 nm. The suspensions were diluted at 1:50 in an RPMI 1640 culture medium to obtain the final concentration of 0.4 × 104 to 5 × 104 CFU/mL.

The inoculated microplates were incubated at 35 °C for 72 h before observing results with the naked eye. The MICs of AMB, ITR, and POS were defined as the lowest concentration that inhibited 100% growth, when compared with an inoculum-free medium control. For the MICs of KET, FLU, MCZ, TERB, and GRI, MICs were defined as the lowest concentration that inhibited 50% growth when compared with a drug-free growth control. The MFCs of all drugs were determined by transferring 200 µL of suspensions of the optically clear well and the higher concentrations (2 to 4 higher concentrations) into the fresh SDB medium in a microtube and incubated at 30 °C for 7 days. The MFCs were detected as the lowest concentration of antifungal drugs that resulted in an optically clear tube by killing all fungal cells. A quality control strain of *Candida parapsilosis* ATCC 22019 was included in the study.

## 3. Results

### 3.1. Geographic Distribution, Demographic Characteristics, and Clinical Data

A total of 42 isolates of *S. schenckii* were phenotypically identified, and 38 cases of feline sporotrichosis were diagnosed in Bangkok and the Bangkok metropolitan region of Thailand. The number of cases, year of diagnosis, and geographic distribution of the sporotrichosis cases are shown in Figure 1. Most cases occurred in 2021 (n = 27) and were predominantly found in the Bangkok area (85%). The other cases were scattered around the surrounding provinces (Nonthaburi: 5%, Pathum Thani: 5%, Ayutthaya: 5%). One zoonotic case was confirmed in 2020 in a cat owner with a history of scratches from her infected cat prior to the occurrence of a skin lesion.

Table 2 summarizes the demographic and clinical characteristics of feline sporotrichosis cases among the Thai cats in this study, including age (mean ± standard deviation [SD]: 3.96 ± 3.93 years), sex (male: 61%, female: 39%), breed (domestic short hair: 95%, purebred: 5%), castration (yes: 58%, no: 42%), cat rearing (outdoor access: 76%, indoor only: 24%), and feline leukemia virus (FeLV)/feline immunodeficiency virus (FIV) infection (no: 55%, yes: 5%, no data: 40%). History of an associated penetrating wound by fighting with other cats was mentioned in 50% of cases. Clinical presentation included skin lesions at different sites (skin lesions at 1 site: 24%, 2 sites: 29%, ≥3: 47%). The lesions mostly appeared as ulcers (92%) and were occasionally found as granulomatous nodules (8%). To treat sporotrichosis, 82% of cases used only antifungal drugs, while the rest involved surgery for wound debridement combined with antifungal drugs. For treatment outcomes, we found that 53% of cases recovered and 37% died; treatment was still ongoing for the other 10% of cases.

### 3.2. Phenotypic Characteristics of S. schenckii Thai Isolates

All isolates included in the phenotypic study (n = 26) were able to grow at 30 and 35 °C. The growth was inhibited at 40 °C, and 31% of isolates were stunted earlier at 37 °C. Three unique colony appearances (mold, yeast-like, and yeast) were consistently exhibited and associated with the culture medium and temperature, as presented in Figure 2. The mold colony was observed as white to brownish aerial mycelia (Figure 2A); the yeast-like colony was white to brownish, with a smooth or wrinkled surface and a fine filamentous edge (Figure 2B); and the yeast colony was white to cream colored, moist, and smooth with a round margin (Figure 2C).

Mold colonies showed a microscopic morphology of hyaline septate hyphae, less differentiated conidiophores, and abundant pigmented, thick-walled, globose to ovoid, and 2.39–5.45 µm long by 1.91–2.96 µm wide sessile conidia (Figure 2D). The yeast-like colonies microscopically revealed septate, smooth, thin-walled, hyaline hyphae, conidiophores not differentiated from vegetative hyphae, and produced abundant clavate to elongate pyriform, which were slightly curved and wider rounded at apex conidia (Figure 2E). The microscopic morphology of the yeast form was similar at all temperatures: 2.39 to 5.64 µm in size and mixed oval and cigar-shaped yeast cells with narrow-based budding (Figure 2F). Some of our isolates (9/26) started producing light brown colonies (Figure 2G), which became heavily dark brown in mold form (Figure 2H) and yeast form (Figure 2I) after 14 days of incubation.

Table 3 shows the progression of the mold, yeast-like, and yeast phases, including the colony sizes in relation to the culture medium and temperature. The mold phase was detected in only 19% of all isolates. The largest colony sizes in the mold phase were observed on SDA and PDA at 30 °C, with diameters of 6.99 to 9.34 mm. The smaller colony sizes presented at 35 °C on SDA and PDA with 4% growth. The yeast-like form, observed as a mix of mycelial and yeast cells, was the most common appearance of *S. schenckii* isolates in our study. This morphology was found in a broad range of growth conditions, especially at 30 °C on SDA/PDA (81%) and BHIA (50%), with a large colony diameter size (3.41 to 4.11 mm), and at 35 ˚C on SDA/PDA (73 to 85%), and with a smaller diameter size (1.74 to 1.97 mm). About 50% of all isolates were able to convert to the yeast phase at 30 °C on BHIA, and the highest conversion percentage was found at 35 °C on BHIA (85%). At higher temperatures (35 and 37 °C), the transition from the mold to the yeast phase occurred on SDA and PDA, but the sizes of the yeast colonies were smaller than those grown on BHIA. The average % GI values at temperatures of 30 to 37 °C were 90.80% on SDA, 95.23% on PDA, and 83.12% on BHIA. Therefore, *S. schenckii* isolates in our study were classified as a low-thermotolerant strain.

### 3.3. Phylogenetic Analysis of S. schenckii Thai Isolates

The sequence of ITS was approximately 655 nucleotides, and all sequences were matched with *S. schenckii* (identity range = 99.83 to 100%). The sequence of CAL was approximately 911 nucleotides, and all strains were matched with *S. schenckii* (identity range = 95.4 to 100%). The sequence of b-TUB was approximately 404 nucleotides, and all sequences were matched with *S. schenckii* (identity range = 99.76 to 100%).

Phylogenetic trees constructed based on the sequences of each gene showed the distribution of a clinical clade of *Sporothrix* spp. into four distinctive species: *S. luriei*, *S. globosa*, *S. brasiliensis*, and *S. schenckii*. All Thai clinical strains were co-clustered with *S. schenckii* strains of different geographic origins, including all reported continents. Based on the ITS sequences, the Thai strains were grouped with clade D of *S. schenckii*, which consists of isolates from Malaysia, the United States, Peru, Argentina, and Brazil (Figure 3). We also aligned the ITS sequences of Thai strains against isolates from Malaysia and other countries. The nucleotide variable sites were detected and showed consistency with the results of the phylogenetic tree analysis. *S. schenckii* clade C and D had a variable site at position 500 (Appendix A). All Thai and Malaysian strains revealed the presence of nucleotide T at position 162, while nucleotide C was noted among other strains in clade D (Appendix A). The single nucleotide polymorphism between the Thai and Malaysian strains was detected once at position 40—read adenine (A) from the Thai strains and thymine (T) from the Malaysian strains (Appendix A).

Based on a phylogenetic analysis of the concatenated CAL gene and b-TUB gene, the *S. schenckii* isolates were divided into five groups according to their location on phylogenetic clades. Group I was a group of clades composed of isolates from various hosts (human and environment) and geographic areas (United States, Japan, and Mexico). Group II was a monophyletic clade of Thai isolates that were mainly from cats, with one from a human. Group III was a clade from South America (Columbia, Peru, Bolivia, and Argentina) and North America (Guatemala), mainly isolated from humans. Group IV was a clade of isolates from Peru, Argentina, and South Africa, mostly from humans. Finally, Group V was a clade of environmental isolates from Mexico (Figure 4).

### 3.4. Antifungal Susceptibility

The in vitro antifungal activities of eight antifungal agents against 26 *S. schenckii* isolates were evaluated based on MIC and MFC. We analyzed the MIC ranges, geometric mean values, the MIC50, and the MIC90 during both the yeast and mold phases (Table 4). For most antifungal drugs, including AMB, ITR, POS, KET, and MCZ, slightly higher MICs (one dilution) were found for the yeast form, but the opposite was true for TERB.

For AMB, MICs ranged from 0.25 to 4 µg/mL and the MIC50 were between 0.5 to 1 µg/mL for the mold and yeast phases, respectively. An MIC of 4 µg/mL was required to inhibit the growth of most isolates (MIC90), and fungicidal activity was observed for MFCs ranging from 0.25 to ≥32 µg/mL and an MFC90 of 16 µg/mL. For ITR and POS, the MIC ranged from 0.25 to ≥32 µg/mL in the yeast phase and 0.0625 to ≥32 µg/mL in the mold phase. The MIC50 were between 0.5 to 1 µg/mL in the yeast phase and 0.25 to 0.5 µg/mL in the mold phase. A higher MIC was required to inhibit the growth of most isolates (MIC90), at 4 µg/mL for ITR and 8 µg/mL for POS. Interestingly, much higher MFCs of ITR and POS, ranging from 0.5 µg/mL to ≥32 µg/mL, were observed for fungicidal activity.

TERB inhibited the growth of isolates at concentrations ranging from 0.0156 to 2 µg/mL in the yeast phase and 0.0312 to 4 µg/mL in the mold phase. The MIC50 concentrations were 0.5 and 0.75 µg/mL for the yeast and mold phases, respectively, and most of our isolates in both phases required an MIC90 of 2 µg/mL to inhibit growth. However, TERB showed fungicidal activity at MFCs ranging from 1 µg/mL to >8 µg/mL, an MFC50 of 8 µg/mL, and an MFC90 of >8 µg/mL. KET and MCZ had the lowest MIC ranges (0.0039 to 1 µg/mL), MIC50 (0.25 µg/mL), and MIC90 (0.5 µg/mL). The efficacy of FLU against isolates in both phases had an MIC ranging from 32 to >256 µg/mL; the MIC50 and MIC90 were >256 µg/mL.

## 4. Discussion

Sporotrichosis is a mycotic infectious disease in humans and animals that is often reported in cats. In this study, we assessed the present outbreak of sporotrichosis in and around Bangkok, Thailand.

Our sample mainly comprised domestic cats that had free access to the surrounding area of their houses in Bangkok. All cases and isolates in our study were obtained from the culture collection of the Microbiology Testing Services of the Department of Microbiology and Immunology, which commonly supports cases from the Kasetsart University Veterinary Teaching Hospital. Therefore, the geographic distribution might not be representative of the feline sporotrichosis outbreak in Bangkok or Thailand. However, we noticed that in 2021, there was a widening distribution of cases from Bangkok to the neighboring provinces to the north, including Nonthaburi, Pathum Thani, and Ayutthaya. A similar geographic distribution was observed in Brazil, which suggested the rapid expansion of zoonotic and feline sporotrichosis cases from the South toward the Northeast [29]. In Thailand, the first case was reported in December 2017 in Bangkok and was followed by 38 cases during 2018–2021 reported in southern Thailand, close to Malaysia, the country with the most reported cases of human and feline sporotrichosis in Southeast Asia [14,24,30]. However, the transboundary expansion of this pathogen from Malaysia to Thailand has not yet been proven.

Most cats with sporotrichosis included in our study were young adults (mean age of 3.96 years), male, and domestic short hairs with uncontrolled outdoor access, similar to previous reports [31,32,33]. Specific demographic characteristics such as age and outdoor access, as well as natural and sexual behaviors of cats, may play a critical role in the transmission of sporotrichosis from cat to cat, cat to dog, and cat to human. An age of 3 years is linked to sexual maturity, and being in heat possibly increased the chance of cats fighting or having contact with other infected animals that bear *Sporothrix* in skin lesions, nails, and oral mucosa [34]. Other cat behaviors that possibly increased their risk of sporotrichosis infection include burying their excrement, climbing trees, or sharpening their claws on trees [35].

The severity of sporotrichosis infections or treatment outcomes were not associated with the underlying presence of FeLV/FIV in our cases, consistent with previous reports [3,36]. In contrast, HIV infection in humans aggravates sporotrichosis, with a higher incidence of severe disseminated cases and a higher number of hospitalizations and deaths [37]. Mortality in our sporotrichosis cases was high among those cats with severe skin lesions—on ≥3 sites of the body (50%)—and those with nasal involvement (21%). Therefore, nasal involvement, respiratory signs, and high fungal loads in skin lesions are prognostic factors of treatment failure in feline sporotrichosis, as suggested by others [38,39]. Other factors may be associated with the failure of treatment, such as low income of the owners, difficulty in administering medication for cats at home, and difficulty in confining cats at home for a long period [33].

Thermotolerance has been known as a factor influencing the virulence of a disease, especially in terms of clinical manifestations. In Brazil, Mexico, and Guatemala where most human cases presented with lymphocutaneous form, strains of *S. brasiliensis* and *S. schenckii* recovered there have shown more resistance to 37 °C [12,40]. A previous report of human sporotrichosis in Malaysia revealed that 7 of 19 cases had a history of cat bites/scratches, and lymphocutaneous sporotrichosis was the most common presentation observed there [41]. We identified a phenotypic trait of high-temperature intolerance and a requirement for lower conversion temperatures in our Thai strains, similar to other Thai and Malaysian strains [24,42]. Therefore, the dissemination via lymphatics or blood circulation or its zoonotic potential was yet a concern among these *S. schenckii s. str.* Thai and Malaysian strains. Furthermore, the low-thermotolerant trait may be related to the common skin lesions in cooler body parts such as ear tips, nasal planum, and nasal cavities in cats [11]. Interestingly, colonization of *Sporothrix* spp. at sites with a lower body temperature, such as the nasal and oral cavities and the nails, in the infected and noninfected cats of endemic areas is suggested to be a potential source of zoonotic transmission [34]. Moreover, melanin contributes to the virulence of fungal agents by protecting them from the host defense response, oxidizing agents, and hydrolytic enzymes, and reduces phagocytosis or the induction of cell death [43]. Some of our isolates started presenting melanized colonies after seven days of incubation and became dark brown pigmented after 14 days of incubation. These relate to *S. brasiliensis*, which is most associated with feline and human sporotrichosis in Brazil; this species has rapid melanization and a high level of pigmentation. On the other hand, *S. globosa*, the most endemic species, caused nearly 99.3% of sporotrichosis in Asia. Most isolates of *S. globasa* were recovered from fixed cutaneous and lymphocutaneous lesions, and this species has never been reported to cause infections in animals. Furthermore, *S. globosa* isolates from Asia and all geographical regions are generally considered less virulent than *S. schenckii s. str.* by presenting very low thermotolerance (97–100% of inability to grow at 37 °C), a low percentage of yeast conversion in lesion and culture, and a low level of melanin production [12,44]. Therefore, further studies on the virulence and pathogenicity of *S. schenckii s. str.*, the most isolated species causing emerging feline sporotrichosis in Southeast Asia with zoonotic potential, should be extended in the future.

Phenotypic identification based on micromorphology, physiology, and biochemical tests is inadequate for the species-level identification of *Sporothrix* spp., which is related to geographic distribution, host range, virulence factors, and antifungal susceptibility [45]. Based on the sequencing of the ITS region of rDNA, all isolates were identified as *S. schenckii s. str.* with a very high identity match (99.83–100%) and were clustered in clinical clade D together with strains originally reported in Malaysia, southern Thailand, and globally. In terms of *S. schenckii s. str.*, intraspecies sequence-based typing has been performed using various methods, such as, for example, mating-type idiomorphs [46], mitochondria DNA (mtDNA) typing [47], and multi-locus sequence analysis (MLSA) of concatenated genes (CAL, b-TUB, ITS, or translation elongation factor [TEF] encoding genes), which is commonly used to determine the genetic relatedness of whole groups of the genus *Sporothrix* [48,49,50]. To demonstrate the geographic relationship of the isolates from Thailand and other geographic origins, a phylogenetic tree was built with the concatenated sequences of the CAL and b-TUB markers. According to the phylogenetic trees, our isolates and feline clinical isolates from another study from the southern part of Thailand were grouped together in the same clade (Group II) with strong bootstrap support (98%), supporting their close genetic relationship and clearly dividing them from others in the phylogenetic tree [24]. There have also been studies of feline and human clinical isolates from a nearby geographic area in Southeast Asia (Malaysia) [23,26]. Even though there is a lack of concordance of gene selection with our analysis, the previous research showed that the Malaysian and Thai *S. schenckii s. str.* isolates were closely related in CAL gene phylogeny, regardless of whether they were from humans or cats [46]. Interestingly, Group IV on the tree was composed of the isolates from different geographic areas (Argentina, Peru, and South Africa), which was the nearest clade from the *S. brasiliensis* outgroup, but this arrangement of clades conformed with previous studies that used two different base methods. One of these methods is amplified fragment length polymorphism, which is a well-acknowledged DNA fingerprint typing method in the *Sporothrix* species [42], and the other is MLSA sequence-based typing of the concatenated chitin synthase (CHS), b-TUB, and CAL genes.

Although the ITS is the most used and most sufficient in distinguishing the species of *Sporothrix*, the bootstrap value of our phylogenetic tree showed that the CAL was the most accurate for clustering the clinically relevant *Sporothrix* spp. These results are supported by previous studies, including Zhang et al. [43], which reported the success rates of multi-locus gene sequencing (ITS, CAL, TEF1, and TEF3) data analysis. They demonstrated that identification with the CAL gene had a high-resolution power by obtaining small intraspecific sequence variability compared with barcoding gaps between species. We suggest using the ITS region for routine identification because the ITS gene is a universal DNA barcode for the identification of fungi species [51]. Using the CAL gene is preferred for epidemiologically sorting *Sporothrix* clinical species.

For antifungal susceptibility testing, we observed higher MICs in the yeast phase, which differed from previous studies [52,53,54] and could possibly be explained by the inoculum sizes used. We followed the CLSI M38-A2 guidelines for both the yeast and mold phases; therefore, the yeast inocula were 40 times higher than the usual recommended use in BMD yeast testing (CLSI M27-A3). Interestingly, consistency in the MIC90s obtained in both phases was detected by following the inoculum size in the CLSI M38-A2. BMD testing against *S. schenckii* is officially described in the CLSI M38-A2 for only the filamentous phase, whereas the infected form in the host body as the yeast phase has not been standardized yet. The correlation between the lower MICs of the yeast phase according to the CLSI M27-A3 of previous studies and the emergence of in vitro resistant strains of *S. schenckii* and in vivo treatment failure is still in question [55]. The MIC breakpoints of *Sporothrix* are not established yet in the CLSI M38-A2 document [28]. Species-specific epidemiological cutoff values (ECVs) for *S. schenckii* and *S. brasiliensis* have been proposed to identify the non-wild type isolate that is equal to a potentially resistant isolate. However, the ECVs for *S. schenckii* were only available for AMB (4 µg/mL), ITR (2 µg/mL), POS (2 µg/mL), and voriconazole (64 µg/mL), and not for KET or TERB yet [56]. In accordance with Espinel-Ingroff et al. [55], we preferred to interpret the antifungal-resistance profile of our isolates using the ECVs versus the MIC, MIC50, and MIC90 values.

For AMB, all our isolates were classified as wild type strains by the MIC50 and MIC90. They were similar to *S. schenckii* isolates found worldwide but different from the non-wild type or AMB-resistant strains that have been identified in Brazil [57]. For ITR, our strains were classified as wild type by the MIC50 but exhibited a non-wild type MIC90 value, similar to the *S. schenckii* from Brazil but different from isolates worldwide that exhibited non-wild type MIC50 and MIC90 values. However, we found four ITR-resistant strains (4/26, 15.38%) with MICs of 4 µg/mL (n = 2) and >32 µg/mL (n = 2). Previous reports of antifungal susceptibilities of *S. schenckii* isolates from southern Thailand showed high resistance to AMB (34.21%) and ITR (26.32%), while studies in Malaysia reported 5% of isolates were ITR-resistant strains [24,42]. ITR is the drug of choice for feline sporotrichosis treatment; however, *S. schenckii* has been found to be less susceptible to ITR than *S. brasiliensis* [58]. In cats, the bioavailability of ITR was observed to be better in oral solution than in tablets or capsules, but it is more expensive and less accessible. Concerns regarding the use of capsules or tablets for treatment in feline sporotrichosis include treatment failure and drug resistance [59]. For POS, our isolates were identified as wild type and non-wild type strains by MIC50 and MIC90, similar to *S. schenckii* reported worldwide [57]. POS is a new triazole drug with fungicidal effects against *Aspergillus* spp., *Candida* spp., and zygomycetes [60]. In our *S. schenckii* isolates, POS was more active than ITR, but POS-resistant and ITR–POS-resistant strains were detected. To the best of our knowledge, POS has never been used in Thai veterinary practices, so the resistance remains questionable.

Because there were ECVs of only four antifungal drugs available for identifying the wild type and non-wild type strains of *S. schenckii*, the susceptibility of other antifungal drugs tested in our strains was compared with the MIC of mode (the most frequent MIC detected in *S. schenckii* worldwide) published by Espinel-Ingroff et al. [55]. Various studies have reported that TERB was the most effective antifungal drug against *S. schenckii*, with low MICs. Data obtained from worldwide testing of 118 isolates had an MIC range of ≤0.03–1 µg/mL, a mode of 0.5 µg/mL (43/118), and the highest reported MIC since 2017 at 1 µg/mL (6/118) [55,61,62,63,64]. Our isolates demonstrated high MICs for TERB at 1 µg/mL (7/26), 2 µg/mL (6/26), and 4 µg/mL (1/26). High MICs for TERB were also reported from cat isolates in Malaysia: 4 µg/mL (23/40 isolates) and 8 µg/mL (4/40 isolates). Since being reported in the 1990s, Malaysian isolates of *S. schenckii s. str.* have exhibited low susceptibility against all antifungal drugs [42]. In our hospital, TERB has been used more frequently, especially in combination with ITR, to treat sporotrichosis cases with low responsiveness. For other less commonly used antifungal drugs, isolates in Thailand were susceptible to KET and MCZ, but the many side effects of KET, such as anorexia, depression, nausea, vomiting, diarrhea, or elevated liver enzymes, are factors to consider before using these drugs [65]. The emerging resistance to commonly used antifungal drugs, including ITR, TERB, and AMB, should be a concern in refractory cases, and antifungal susceptibility testing might be required.

Cat-transmitted sporotrichosis has garnered worldwide public concern, and a One Health approach has been implemented in South American countries to control and prevent the spread of the disease in animals, humans, and the environment. Veterinarians play an important role in controlling this disease, including investigating and addressing the outbreak; educating cat owners on the importance and difficulties of treatment, management, and control of their infected cat; understanding the risks and sources of zoonosis; and decontaminating and responsibly handling deceased cats [66]. Since 2018, Thailand has faced a feline sporotrichosis outbreak. However, the true burden of this disease in cats and humans has not been investigated. Further studies and programs focused on epidemiology, surveillance, and controlling the spread of sporotrichosis among the cat population may help reduce the risk of cat-transmitted sporotrichosis to other animals and humans.

## 5. Conclusions

We described 38 cases in the recent feline sporotrichosis outbreak caused by *S. schenckii s. str.* in Bangkok and the surrounding provinces, which are in the central part of Thailand. Our Thai isolates reported here share phenotypic and genotypic characteristics with those distributed in Malaysia and southern Thailand. Susceptibility testing of the Thai isolates showed that AMB, ITR, and POS MIC50s were within the limit of ECVs and compatible with the wild type traits. However, high MICs and MFCs of ITR and POS have been detected in some non-wild type or resistant strains. The MICs of TERB observed within our isolates were higher than those frequently reported worldwide. We recommend that veterinarians perform fungal cultures in every possible case and reserve antifungal drug susceptibility tests for cats that are unresponsive to antifungal drugs. This approach would be not only beneficial for treatment outcomes but also essential for controlling and preventing the spread of disease from infected cats to other animals and humans in Thailand.

## Figures and Tables

**Figure 1 jof-09-00590-f001:**
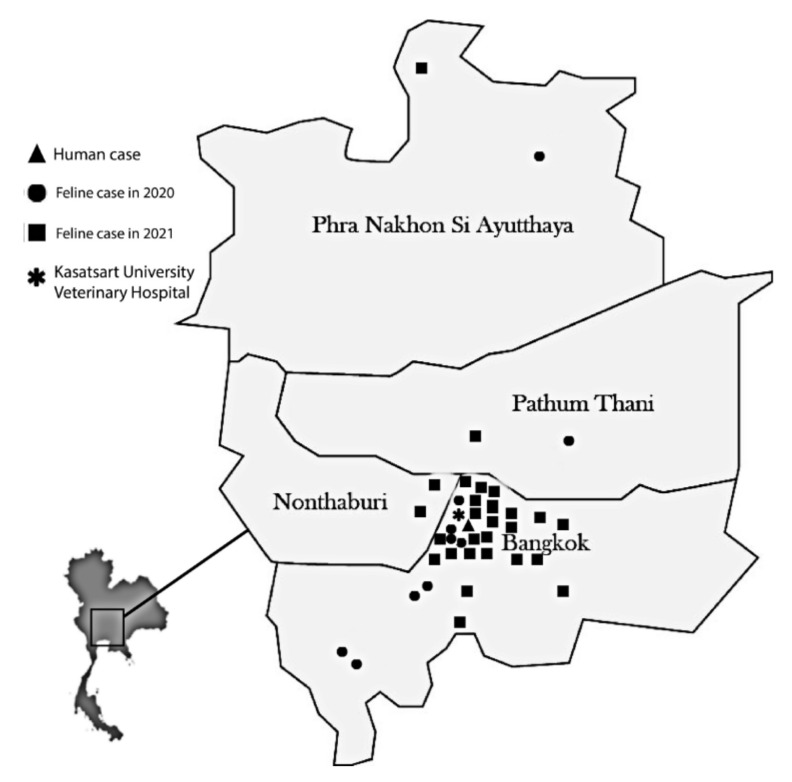
Geographic distribution of 38 cats with sporotrichosis in Bangkok and the Bangkok metropolitan regions of Thailand during 2020 and 2021. Each data point represents a single case in a given year.

**Figure 2 jof-09-00590-f002:**
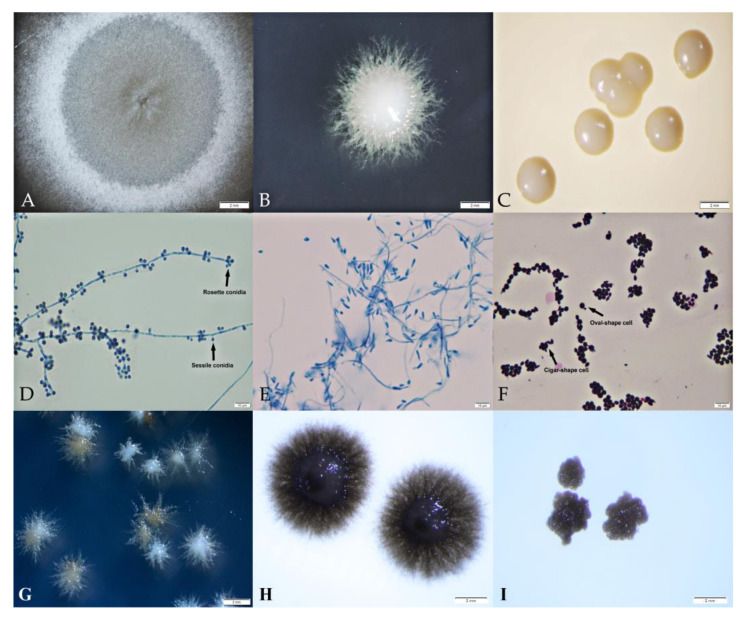
Features of *S. schenckii* colonies in this study incubated at 30 and 35 °C for 14 days. The colonies exhibited different morphologies: the mold form was white aerial mycelia (**A**), the yeast-like form had a smooth surface at the center with a filamentous edge (**B**), and the yeast form presented as white to cream-colored, with a smooth surface and round margin (**C**). Microscopic morphologies are also shown in the rosette and sessile conidia (**D**), the yeast-like cell and filament (**E**), and the yeast cells (**F**). After 7 days of incubation, some isolates were observed to have a light brown color (**G**), and for 14 days, the isolates turned into dark brown colors in mold form (**H**) and yeast form (**I**).

**Figure 3 jof-09-00590-f003:**
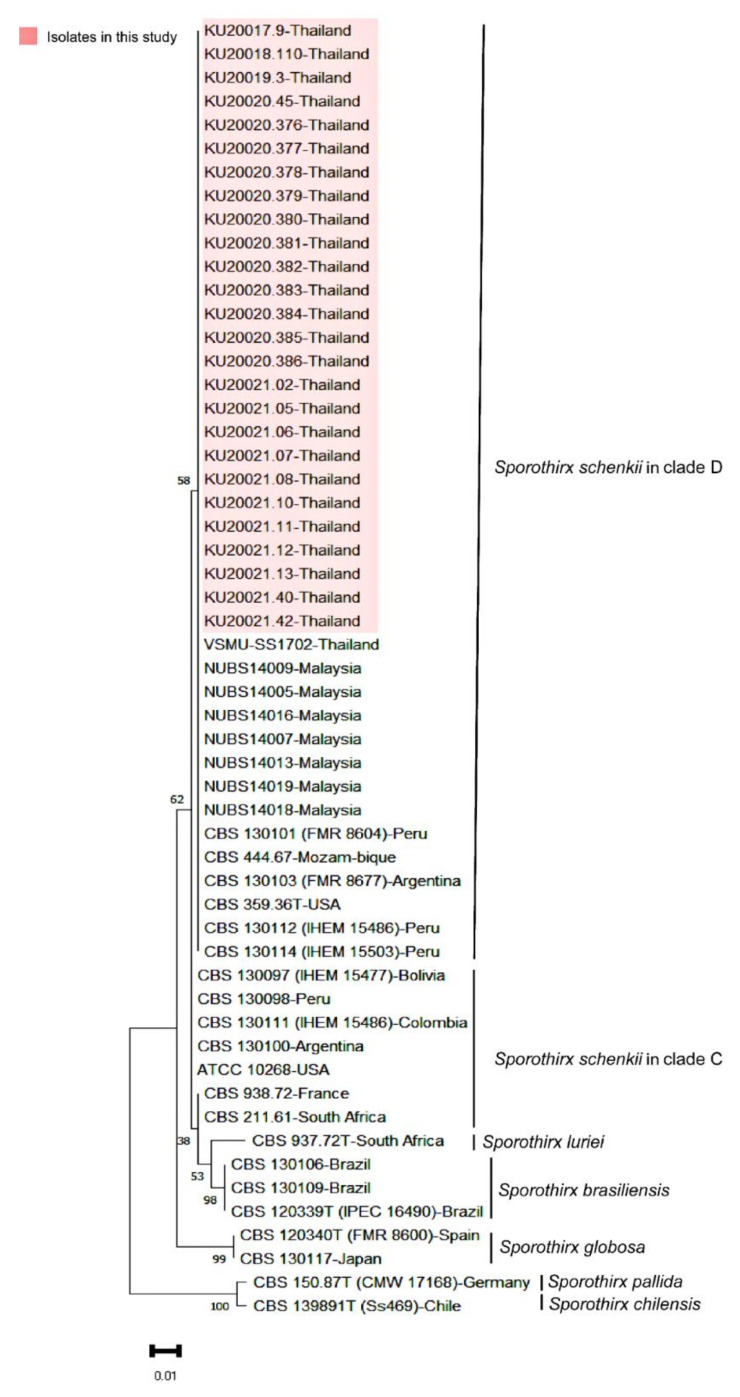
Phylogenetic analysis of the ITS region of rDNA *Sporothrix* spp. Sequences obtained during this study are presented in the orange box. The species, country, and strain are indicated to the right of the sequence. Evolutionary analyses were conducted in MEGA11, using 1000 bootstrap replicates.

**Figure 4 jof-09-00590-f004:**
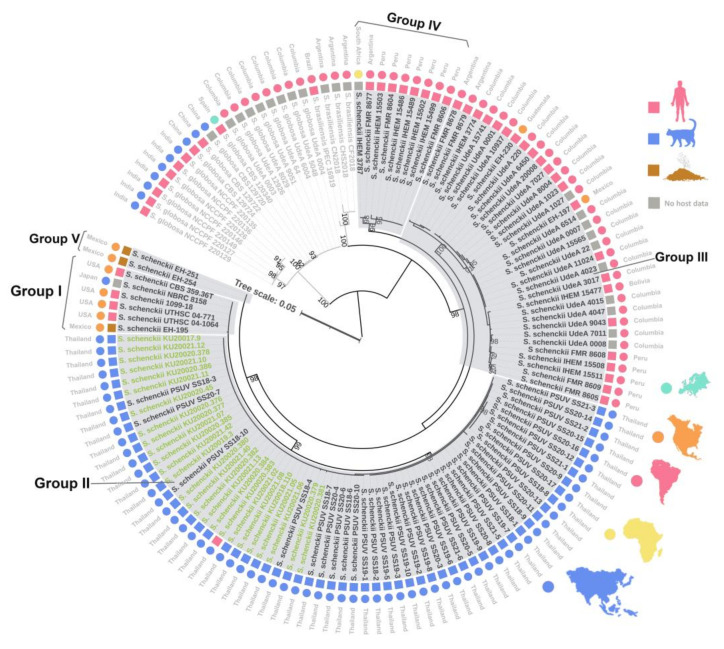
Phylogenetic (ML) analysis, based on the concatenated calmodulin and beta-tubulin genes of *S. schenckii* plus 4 *S. brasiliensis* and 19 *S. globosa* sequences as the closely related outgroup. The color of the squares indicates the host of the isolates, and the color of the circles shows the continent of origin. The strain names in green color represented strains isolated in this study.

**Table 1 jof-09-00590-t001:** *Sporothrix* spp. isolates used for phylogenetic analysis in this study.

Species	Strain	Source	Origin	Accession Number		
				ITS	CAL	b-TUB
*S. schenckii*	KU20017.9	Cat	Thailand	**MW819850**	**MZ215901**	**MZ215921**
*S. schenckii*	KU2001810	Cat	Thailand	**MW819852**	**MZ215902**	**MZ215922**
*S. schenckii*	KU20019.3	Cat	Thailand	**MW819851**	**MZ215903**	**MZ215923**
*S. schenckii*	KU20020.45	Cat	Thailand	**MW857103**	**MZ215906**	**MZ215926**
*S. schenckii*	KU20020.376	Cat	Thailand	**MW857105**	**MZ215907**	**MZ215927**
*S. schenckii*	KU20020.377	Cat	Thailand	**MW857106**	**MZ215908**	**MZ215928**
*S. schenckii*	KU20020.378	Cat	Thailand	**MW857108**	**MZ215909**	**MZ215929**
*S. schenckii*	KU20020.379	Cat	Thailand	**MW857109**	**MZ215910**	**MZ215930**
*S. schenckii*	KU20020.380	Cat	Thailand	**MW857111**	**MZ215911**	**MZ215931**
*S. schenckii*	KU20020.381	Cat	Thailand	**MW857112**	**MZ215912**	**MZ215932**
*S. schenckii*	KU20020.382	Cat	Thailand	**MW857113**	**MZ215913**	**MZ215933**
*S. schenckii*	KU20020.383	Human	Thailand	**MW857114**	**MZ215914**	**MZ215934**
*S. schenckii*	KU20020.384	Cat	Thailand	**MW857115**	**MZ215915**	**MZ215935**
*S. schenckii*	KU20020.385	Cat	Thailand	**MW857126**	**MZ215916**	**MZ215936**
*S. schenckii*	KU20020.386	Cat	Thailand	**MW857128**	**MZ215917**	**MZ215937**
*S. schenckii*	KU20021.02	Cat	Thailand	**MW857129**	**MZ215919**	**MZ215939**
*S. schenckii*	KU20021.05	Cat	Thailand	**MZ882155**	**OK065908**	**OK065918**
*S. schenckii*	KU20021.06	Cat	Thailand	**MZ882257**	**OK065909**	**OK065919**
*S. schenckii*	KU20021.07	Cat	Thailand	**MZ882288**	**OK065910**	**OK065920**
*S. schenckii*	KU20021.08	Cat	Thailand	**MZ882289**	**OK065911**	**OK065921**
*S. schenckii*	KU20021.10	Cat	Thailand	**MZ882290**	**OK065912**	**OK065922**
*S. schenckii*	KU20021.11	Cat	Thailand	**MZ882291**	**OK065913**	**OK065923**
*S. schenckii*	KU20021.12	Cat	Thailand	**MZ882292**	**OK065914**	**OK065924**
*S. schenckii*	KU20021.13	Cat	Thailand	**MZ882293**	**OK065915**	**OK065925**
*S. schenckii*	KU20021.40	Cat	Thailand	**MZ882294**	**OK065916**	**OK065926**
*S. schenckii*	KU20021.42	Cat	Thailand	**MZ882295**	**OK065917**	**OK065927**
*S. schenckii*	UTHSC 04-1064	Human	USA	-	AM399014	AM747285
*S. schenckii*	UTHSC 04-771	Human	USA	-	AM398985	AM747287
*S. schenckii*	UdeA_0001	Human	Columbia	-	ON060845	MG953922
*S. schenckii*	UdeA_0007	ND	Columbia	-	ON060848	ON023043
*S. schenckii*	UdeA_0008	ND	Columbia	-	ON060849	ON023044
*S. schenckii*	UdeA_22	ND	Columbia	-	ON060860	MG953936
*S. schenckii*	UdeA_220	Human	Columbia	-	ON060861	MG953935
*S. schenckii*	UdeA 1023	Human	Columbia	-	ON060850	MG953933
*S. schenckii*	UdeA 1027	ND	Columbia	-	ON060851	MG953934
*S. schenckii*	UdeA_3017	Human	Columbia	-	ON060863	MG953916
*S. schenckii*	UdeA_4015	ND	Columbia	-	ON060864	ON023050
*S. schenckii*	UdeA_4023	ND	Columbia	-	ON060865	MG953917
*S. schenckii*	UdeA_4047	ND	Columbia	-	ON060866	ON023051
*S. schenckii*	UdeA_6450	Human	Columbia	-	ON060870	MG953918
*S. schenckii*	UdeA_6514	ND	Columbia	-	ON060871	ON023052
*S. schenckii*	UdeA_7011	ND	Columbia	-	ON060872	ON023053
*S. schenckii*	UdeA_7027	Human	Columbia	-	ON060873	MH174665
*S. schenckii*	UdeA_8004	Human	Columbia	-	ON060874	MG953919
*S. schenckii*	UdeA_9043	Human	Columbia	-	ON060876	MH174668
*S. schenckii*	UdeA 10937	Human	Columbia	-	ON060852	MG953927
*S. schenckii*	UdeA 11024	Human	Columbia	-	ON060853	MG953925
*S. schenckii*	UdeA 15565	Human	Columbia	-	ON060857	MG953924
*S. schenckii*	UdeA 15741	Human	Columbia	-	ON060858	MG953920
*S. schenckii*	UdeA 20008	Human	Columbia	-	ON060859	MG953921
*S. schenckii*	PSUV_SS18-1	Cat	Thailand	-	ON211523	ON211485
*S. schenckii*	PSUV_SS18-2	Cat	Thailand	-	ON211527	ON211489
*S. schenckii*	PSUV_SS18-3	Cat	Thailand	-	ON211510	ON211472
*S. schenckii*	PSUV_SS18-4	Cat	Thailand	-	ON211508	ON211470
*S. schenckii*	PSUV_SS18-6	Cat	Thailand	-	ON211517	ON211479
*S. schenckii*	PSUV_SS18-7	Cat	Thailand	-	ON211511	ON211473
*S. schenckii*	PSUV_SS18-8	Cat	Thailand	-	ON211520	ON211482
*S. schenckii*	PSUV_SS18-9	Cat	Thailand	-	ON211519	ON211481
*S. schenckii*	PSUV_SS18-10	Cat	Thailand	-	ON211529	ON211491
*S. schenckii*	PSUV_SS19-1	Cat	Thailand	-	ON211515	ON211477
*S. schenckii*	PSUV_SS19-2	Cat	Thailand	-	ON211513	ON211475
*S. schenckii*	PSUV_SS19-3	Cat	Thailand	-	ON211514	ON211476
*S. schenckii*	PSUV_SS19-5	Cat	Thailand	-	ON211542	ON211504
*S. schenckii*	PSUV_SS19-6	Cat	Thailand	-	ON211530	ON211492
*S. schenckii*	PSUV_SS19-7	Cat	Thailand	-	ON211540	ON211502
*S. schenckii*	PSUV_SS19-8	Cat	Thailand	-	ON211524	ON211486
*S. schenckii*	PSUV_SS19-9	Cat	Thailand	-	ON211518	ON211480
*S. schenckii*	PSUV_SS19-10	Cat	Thailand	-	ON211529	ON211491
*S. schenckii*	PSUV_SS20-3	Cat	Thailand	-	ON211516	ON211478
*S. schenckii*	PSUV_SS20-4	Cat	Thailand	-	ON211543	ON211505
*S. schenckii*	PSUV_SS20-5	Cat	Thailand	-	ON211525	ON211487
*S. schenckii*	PSUV_SS20-6	Cat	Thailand	-	ON211534	ON211496
*S. schenckii*	PSUV_SS20-7	Cat	Thailand	-	ON211539	ON211501
*S. schenckii*	PSUV_SS20-8	Cat	Thailand	-	ON211521	ON211483
*S. schenckii*	PSUV_SS20-9	Cat	Thailand	-	ON211522	ON211484
*S. schenckii*	PSUV_SS21-1	Cat	Thailand	-	ON211535	ON211497
*S. schenckii*	PSUV_SS21-2	Cat	Thailand	-	ON211537	ON211499
*S. schenckii*	PSUV_SS21-3	Cat	Thailand	-	ON211538	ON211500
*S. schenckii*	PSUV_SS21-5	Cat	Thailand	-	ON211544	ON211506
*S. schenckii*	PSUV_SS21-6	Cat	Thailand	-	ON211545	ON211507
*S. schenckii*	PSUV_SS20-10	Cat	Thailand	-	ON211541	ON211503
*S. schenckii*	PSUV_SS20-11	Cat	Thailand	-	ON211526	ON211488
*S. schenckii*	PSUV_SS20-12	Cat	Thailand	-	ON211528	ON211490
*S. schenckii*	PSUV_SS20-13	Cat	Thailand	-	ON211512	ON211474
*S. schenckii*	PSUV_SS20-14	Cat	Thailand	-	ON211531	ON211493
*S. schenckii*	PSUV_SS20-15	Cat	Thailand	-	ON211533	ON211495
*S. schenckii*	PSUV_SS20-16	Cat	Thailand	-	ON211532	ON211494
*S. schenckii*	PSUV_SS20-17	Cat	Thailand	-	ON211536	ON211498
*S. schenckii*	NBRC_8158	ND	Japan	-	AM117438	AM116910
*S. schenckii*	IHEM_3774	Human	Colombia	-	AM117447	AM116921
*S. schenckii*	IHEM_3787	ND	South Africa	-	AM117435	AM116925
*S. schenckii*	IHEM_15502	Human	Peru	-	AM117427	AM116927
*S. schenckii*	IHEM_15503	Human	Peru	-	AM117433	AM116930
*S. schenckii*	IHEM_15508	Human	Peru	-	AM117443	AM116924
*S. schenckii*	IHEM_15511	Human	Peru	-	AM117440	AM116917
*S. schenckii*	IHEM_15477	Human	Bolivia	-	AM117444	AM116916
*S. schenckii*	IHEM_15486	Human	Peru	-	AM117432	AM116929
*S. schenckii*	IHEM_15489	Human	Peru	-	AM117430	AM116926
*S. schenckii*	IHEM_15499	Human	Peru	-	AM117434	AM116928
*S. schenckii*	FMR_8604	Human	Peru	-	AM117429	AM116914
*S. schenckii*	FMR_8605	Human	Peru	-	AM117442	AM116923
*S. schenckii*	FMR_8606	Human	Peru	-	AM117431	AM116913
*S. schenckii*	FMR_8608	Human	Peru	-	AM117441	AM116919
*S. schenckii*	FMR_8609	Human	Peru	-	AM117439	AM116918
*S. schenckii*	FMR_8677	Human	Argentina	-	AM117436	AM116915
*S. schenckii*	FMR_8678	Human	Argentina	-	AM117446	AM116920
*S. schenckii*	FMR_8679	Human	Argentina	-	AM117445	AM116922
*S. schenckii*	EH-195	ENV	Mexico	-	ON060840	ON023042
*S. schenckii*	EH-197	Human	Mexico	-	ON060841	MH174664
*S. schenckii*	EH-230	Human	Guatemala	-	ON060842	MG953930
*S. schenckii*	EH-251	ENV	Mexico	-	ON060843	MG953938
*S. schenckii*	EH-254	ENV	Mexico	-	ON060844	MG953937
*S. schenckii*	1099-18	Human	USA	-	JF313360	AXCR01000005
*S. schenckii*	VSMU-SS1702	Human	Thailand	MG270182	-	-
*S. schenckii*	NUBS14005	Cat	Malaysia	LC012513	-	-
*S. schenckii*	NUBS14007	Cat	Malaysia	LC012518	-	-
*S. schenckii*	NUBS14009	Cat	Malaysia	LC012520	-	-
*S. schenckii*	NUBS14013	Cat	Malaysia	LC012599	-	-
*S. schenckii*	NUBS14016	Cat	Malaysia	LC012602	-	-
*S. schenckii*	NUBS14018	Cat	Malaysia	LC012604	-	-
*S. schenckii*	NUBS14019	Cat	Malaysia	LC012605	-	-
*S. schenckii*	FMR 8609	Human	Peru	-	AM117439	AM116918
*S. schenckii*	CBS 359.36^T^	Human	USA	KP017100	AM117437	AM116911
*S. schenckii*	CBS 444.67	ND	Mozambique	KP017099	-	-
*S. schenckii*	CBS 130097	Human	Bolivia	KC113218	-	-
*S. schenckii*	CBS 130098	Human	Peru	KP017091	-	-
*S. schenckii*	ATCC 10268	Human	USA	AB122038	-	-
*S. schenckii*	CBS 130111	Human	Colombia	KC113219	-	-
*S. schenckii*	CBS 130103	Human	Argentina	KC113222	-	-
*S. schenckii*	CBS 130114	Human	Peru	KC113221	-	-
*S. schenckii*	CBS 130101	Human	Peru	KP017095	-	-
*S. schenckii*	CBS 130112	Human	Peru	KC113223	-	-
*S. schenckii*	CBS 211.61	ND	South Africa	KP017093	-	-
*S. schenckii*	CBS 938.72	Human	France	KP017094	-	-
*S. schenckii*	CBS 130100	Human	Argentina	KC113217	-	-
*S. globosa*	CBS 130117	Human	Japan	KC113229	-	-
*S. globosa*	CBS 120340 ^T^	Human	Spain	KC113229	KP101459.	WOEH00000000
*S. globosa*	CBS 129720	ND	China	-	KP101463	KC113237
*S. globosa*	CBS 129722	Human	China	-	KP101464	KC113236
*S. globosa*	CBS 129724	Human	China	-	KP101466	KC113238
*S. globosa*	UdeA 0003	ND	Columbia	-	ON060846	MG953948
*S. globosa*	UdeA 0004	ND	Columbia	-	ON060847	MG953945
*S. globosa*	UdeA 64	ND	Columbia	-	ON060869	MG953946
*S. globosa*	UdeA 5048	ND	Columbia	-	ON060867	MG953939
*S. globosa*	UdeA 6004	ND	Columbia	-	ON060868	MG953940
*S. globosa*	UdeA 8029	ND	Columbia	-	ON060875	MH174666
*S. globosa*	UdeA 9051	ND	Columbia	-	ON060877	MG953941
*S. globosa*	UdeA 12926	ND	Columbia	-	ON060855	MH174667
*S. globosa*	NCCPF 220126	Human	India	-	MN243809	KX881724
*S. globosa*	NCCPF 220127	Human	India	-	MN243810	KX881725
*S. globosa*	NCCPF 220129	Human	India	-	MN243811	KX881701
*S. globosa*	NCCPF 220135	Human	India	-	MN243812	KX881721
*S. globosa*	NCCPF 220136	Human	India	-	MN243813	KX881722
*S. globosa*	NCCPF 220146	Human	India	-	MN243822	KX881719
*S. globosa*	NCCPF 220149	Human	India	-	MN243823	MN257636
*S. luriei*	CBS 937.72 ^T^	Human	South Africa	AB128012	-	-
*S. brasiliensis*	CBS 120339 ^T^	Human	Brazil	KP017087	-	-
*S. brasiliensis*	CBS 130106	Human	Brazil	KC113212	-	-
*S. brasiliensis*	CBS 130109	Human	Brazil	KC113213	-	-
*S. brasiliensis*	CF2018	ND	Argentina	-	MK850446	MK850450
*S. brasiliensis*	CHS2018	ND	Argentina	-	MK850448	MK850452
*S. brasiliensis*	CN2018	ND	Argentina	-	MK850447	MK850451
*S. brasiliensis*	IPEC_16919	Human	Brazil	-	AM116898	AM116934
*S. pallida*	CBS 150.87 ^T^	ENV	Germany	EF127879	-	-
*S. chilensis*	CBS 139891 ^T^	Human	Chile	KP711811	-	-

Sequences obtained during this study are indicated in bold. b-TUB: Beta-tubulin; CAL: Calmodulin; ENV: environment; ITS: internal transcribed spacer region of the nuclear ribosomal DNA gene; ND: not defined; ^T^: Type strain of species.

**Table 2 jof-09-00590-t002:** Geographic distribution, demographic characteristics, and clinical data.

Characteristic		Number (%)
**Case number**		38
**Geographic origin**	Bangkok	32 (85)
	Nonthaburi	2 (5)
	Pathum Thani	2 (5)
	Ayutthaya	2 (5)
**Age (years)**	mean (±standard deviation)	3.96 (±3.93)
	Youngest	6 M
	Oldest	14 Y 10 M
**Sex**	Male	23 (61)
	Female	15 (39)
**Breed**	Domestic short hair	36 (95)
	Purebred	2 (5)
**Castration**	Yes	22 (58)
	No	16 (42)
**Cat rearing**	Indoor only	9 (24)
	Outdoor access	29 (76)
**Possible transmission**	Fights with other cats	19 (50)
	None	11 (29)
	Unknown	8 (21)
**FeLV/FIV infected**	Yes	2 (5)
	No	21 (55)
	No data	15 (40)
**Number of lesions**	1	9 (24)
	2	11 (29)
	≥3	18 (47)
**The form of lesion**	Ulcer	35 (92)
	Granulomatous nodular	3 (8)
**Other organ involvement**	Respiratory system	8 (21)
	Eyes lids	2 (5)
	Bone	1 (3)
	None	27 (71)
**Therapeutics**	Antifungal drugs	31 (82)
	Combined surgy and anti-fungal drugs	7 (18)
**Outcome**	Recovery	20 (53)
	Stable disease and continue treatment	4 (10)
	death	14 (37)

FeLV: feline leukemia virus; FIV: feline immunodeficiency virus; M: month; Y: year.

**Table 3 jof-09-00590-t003:** Average colony diameter (mm) after 14 days of incubation and number (%) of *S. schenckii* Thai strains by culture medium and temperature.

Growth Condition	Growth (Number, %) and Average Colony Diameter (mm)			
	No Growth	Mold	Yeast-Like	Yeast
30 °C				
SDA	0	5, 19 (6.99)	21, 81 (3.41)	0
PDA	0	5, 19 (9.34)	21, 81 (4.11)	0
BHIA	0	0	13, 50 (4.11)	13, 50 (4.15)
35 °C				
SDA	0	1, 4 (0.74)	22, 85 (1.74)	3, 12 (2.24)
PDA	0	1, 4 (1.77)	19, 73 (1.97)	6, 23 (2.15)
BHIA	0	0	4, 15 (3.91)	22, 85 (3.33)
37 °C				
SDA	8 (31)	0	6, 23 (0.39)	12, 46 (0.48)
PDA		0	8, 31 (0.18)	10, 38 (0.38)
BHIA		0	3, 11 (0.46)	15, 58 (0.98)

The numbers in the table indicate the number and percentage of isolates. The values in parentheses are average colony diameter (mm). BHIA: brain heart infusion agar; PDA: potato dextrose agar; SDA: Sabouraud dextrose agar.

**Table 4 jof-09-00590-t004:** Antifungal susceptibility of 26 *S. schenckii* Thai strains in both yeast and mold forms.

Agents	Yeast Phase				Mold Phase			
	MIC Range	Mode	MIC50	MIC90	MIC Range	Mode	MIC50	MIC90
AMB	0.25 to 4	2	1	4	0.25 to 4	0.5	0.5	4
ITR	0.25 to >32	1	1	4	0.0625 to ≥32	0.5	0.5	4
POS	0.25 to >32	0.5	0.5	8	0.0625 to ≥32	0.25	0.25	8
TERB	0.0156 to 2	1	0.5	2	0.0312 to 4	1	0.75	2
KET	0.0039 to 1	0.5	0.25	0.5	0.0039 to 1	0.25	0.25	0.5
MCZ	0.0039 to 1	0.5	0.25	0.5	0.0039 to 1	0.25	0.25	0.5
FLU	32 to >256	>256	>256	>256	32 to >256	>256	>256	>256
GRI	0.0625 to ≥32	4	4	32	0.0625 to ≥32	4	4	16

AMB: amphotericin B; FLU: fluconazole; GRI: griseofulvin; ITR: itraconazole; KET: ketoconazole; MCZ: miconazole; MIC: minimum inhibitory concentration; POS: posaconazole; TERB: terbinafine. Mode is the MIC that represents the most frequently obtained MIC. MIC50 and MIC90 values were recognized as the minimum concentrations of antifungal agents at which 50% and 90% of the growth of *S. schenckii* strains were inhibited.

## Data Availability

The data presented in this study are available within the article and Appendix A.

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
