# Peer review of "Phenotypic and Genotypic Characterization and Antifungal Susceptibility of Sporothrix schenckii sensu stricto Isolated from a Feline Sporotrichosis Outbreak in Bangkok, Thailand"

_jof, 2023, doi:10.3390/jof9050590_

Round 1
Reviewer 1 Report
The manuscript "Phenotypic and Genotypic Characterization and Antifungal Susceptibility of Sporothrix Schenckii Sensu Stricto Isolated from a Feline Sporotrichosis Outbreak in Bangkok, Thailand" is an epidemiologically interesting, well-structured paper, with a methodology consistent with its objectives. The authors present the pheno and genotypic characterization of Sporothrix Schenckii isolates obtained from cats, during an outbreak of feline sporotrichosis, and also evaluated their susceptibility to antifungals. However, I do have some comments.
Materials and methods
Section “2.1. Fungal Isolates and Data Collection”
The authors mention the inclusion of 42 isolates: cats (n=38) and a cat owner (n=1) and three isolates from untraceable source data (n=3), however, the pheno and genotypic characterization was only performed with 26. isolated and do not justify what happened to the rest of the isolates.
Section “2.3. Genotypic Characterization”
The authors obtained the sequences of their 26 isolates, using the three markers (ITS, CAL and bTUB), however, they do not explain why the phylogenetic analysis was carried out separately, that is, one analysis using the ITS sequences and another with the concatenated sequences of b-TUB and CAL. Would it be possible to carry out a phylogenetic analysis with the concatenated sequences of the three markers? and include a greater number of reference sequences of S. globosa?, since in the literature it is suggested that S. globosa is an endemic species of Asia.
Results
In the section “3.3. Phylogenetic Analysis of S. schenckii Thai Isolates”, the authors mention the identity ranges (identity range %) of the sequences obtained with the ITS and CAL markers, however, they do not mention the values obtained with the b-TUB marker.
The quality of figure 4 should be improved, since the names of the isolates are not appreciated, as well as the bootstrap values. In addition, I suggest highlighting in the tree the 26 isolates included in this study.
Why didn't you carry out a phylogenetic analysis with the concatenated sequences of the three markers (ITS, CAL and b-TUB)?
Discussion
Some authors suggest that S. globosa is an endemic species of Asia, since a prevalence of 99.3% has been evidenced, however in this work the authors identified their isolates as Sporothrix schenckii Sensu Stricto, it would be convenient to discuss this finding.
Minor comments
Lines 126-128: I suggest correcting, since the authors mention (Species of all isolates were identified using the BLASTn .....), however this statement is not correct, since BLAST is an algorithm and computer program to align problem sequences with a database and find the sequences most similar to the problem sequence, however, it is a heuristic algorithm, which does not guarantee that the correct solution will be found, so it is a tool that only makes a prediction of the problem sequences and it is necessary to carry out a phylogenetic analysis to obtain a more precise result of the identification.
Line 209: Change “S. Schenckii” by “S. schenckii”
References
Please check references carefully and strictly follow the journal format.
Reviewer 2 Report
In the manuscript, number jof-2368986, entitled: “Phenotypic and genotypic characterization and antifungal susceptibility of Sporothrix schenckii sensu stricto isolated from a feline sporotrichosis outbreak in Bangkok, Thailand”, Yingchanakiat et al. characterize 38 feline sporotrichosis cases in and around Bangkok, Thailand (from 2017 to 2021). This type of studies could help to control the spread of sporotrichosis and reduce the risk ofthis disease transmitted from cats to humans.
Here are the review points:
1) The term S. schenckii complex is not well used, please review and modify.
2) Please, move Table 1 to supplementary material.
3) In the section 3.1, according to previously described, only 26 isolates of Sporotrix spp. were characterized and not 38 as mentioned.
4) In Figure 2, indicate the scale in the photos (in µm).
5) Include information about the different clinical isolates of Sporothrix spp. formed melanin, since it is an important virulence factor of the fungus. Please, include photos.
Minor editing of English language required.
Round 2
Reviewer 1 Report
Comments
I thank the authors of the manuscript “Phenotypic and Genotypic Characterization and Antifungal Susceptibility of Sporothrix Schenckii Sensu Stricto Isolated from a Feline Sporotrichosis Outbreak in Bangkok, Thailand”, for taking my comments into account. I consider that the work has improved, however, I have some comments.
Given the contributions of the work to the epidemiology of sporotrichosis and the impact factor of the journal, it is very important that the sequences obtained from its isolates, with the three markers used (b-TUB, CAL and ITS) be deposited in the GenBank before being published and the access numbers should be included in the manuscript.
On the other hand, it would be convenient to homogenize the nomenclature of the groupings in the two trees, unless there is a justification for it to remain unchanged, since the tree built with the sequences obtained with the ITS marker uses the term "clades" to the groupings, while in the tree built with the concatenated sequences obtained with the b-TUB and CAL markers, they use the term “clusters”, which may confuse readers.
Line 420: I suggest replacing the paragraph “In this study, ML phylogenetic trees based on concatenated CAL and b-TUB genes were used to determine genetic relatedness of S. schenckii s. str. species.” by the paragraph "To demonstrate the geographic relationship of the isolates from Thailand and Malaysia and other geographic origins, a phylogenetic tree was built with the concatenated sequences of the CAL and b-TUB markers"
Author Response
"Please see the attachment."

Reviewer 2 Report
I have no observations.
Author Response
Thank you for reviewer's consideration